# Hierarchical Density Order Embeddings

**Ben Athiwaratkun, Andrew Gordon Wilson**
Cornell University
Ithaca, NY 14850, USA

## Abstract

By representing words with probability densities rather than point vectors, probabilistic word embeddings can capture rich and interpretable semantic information and uncertainty (Vilnis & McCallum, 2014; Athiwaratkun & Wilson, 2017). The uncertainty information can be particularly meaningful in capturing *entailment* relationships – whereby general words such as "entity" correspond to broad distributions that encompass more specific words such as "animal" or "instrument". We introduce *density order embeddings*, which learn hierarchical representations through encapsulation of probability distributions. In particular, we propose simple yet effective loss functions and distance metrics, as well as graph-based schemes to select negative samples to better learn hierarchical probabilistic representations. Our approach provides state-of-the-art performance on the WordNet hypernym relationship prediction task and the challenging HyperLex lexical entailment dataset – while retaining a rich and interpretable probabilistic representation.

## 1 Introduction

Learning feature representations of natural data such as text and images has become increasingly important for understanding real-world concepts. These representations are useful for many tasks, ranging from semantic understanding of words and sentences (Mikolov et al., 2013; Kiros et al., 2015), image caption generation (Vinyals et al., 2015), textual entailment prediction (Rocktäschel et al., 2015), to language communication with robots (Bisk et al., 2016).

Meaningful representations of text and images capture visual-semantic information, such as hierarchical structure where certain entities are abstractions of others. For instance, an image caption "A dog and a frisbee" is an abstraction of many images with possible lower-level details such as a dog jumping to catch a frisbee or a dog sitting with a frisbee (Figure 1a). A general word such as "object" is also an abstraction of more specific words such as "house" or "pool". Recent work by Vendrov et al. (2015) proposes learning such asymmetric relationships with *order embeddings* – vector representations of non-negative coordinates with partial order structure. These embeddings are shown to be effective for word hypernym classification, image-caption ranking and textual entailment (Vendrov et al., 2015).

Another recent line of work uses probability distributions as rich feature representations that can capture the semantics and uncertainties of concepts, such as Gaussian word embeddings (Vilnis & McCallum, 2014), or extract multiple meanings via multimodal densities (Athiwaratkun & Wilson, 2017). Probability distributions are also natural at capturing orders and are suitable for tasks that involve hierarchical structures. An abstract entity such as "animal" that can represent specific entities such as "insect", "dog", "bird" corresponds to a broad distribution, encapsulating the distributions for these specific entities. For example, in Figure 1c, the distribution for "insect" is more concentrated than for "animal", with a high density occupying a small volume in space.

Such entailment patterns can be observed from density word embeddings through *unsupervised* training based on word contexts (Vilnis & McCallum, 2014; Athiwaratkun & Wilson, 2017). In the unsupervised settings, density embeddings are learned via maximizing the similarity scores between nearby words. The density encapsulation behavior arises due to the word occurrence pattern that a general word can often substitute more specific words; for instance, the word "tea" in a sentence "I like iced tea" can be substituted by "beverages", yielding another natural sentence "I like iced beverages". Therefore, the probability density of a general concept such as "beverages" tends to have

a larger variance than specific ones such as "tea", reflecting higher uncertainty in meanings since a general word can be used in many contexts. However, the information from word occurrences alone is not sufficient to train meaningful embeddings of some concepts. For instance, it is fairly common to observe sentences "Look at the cat", or "Look at the dog", but not "Look at the mammal". Therefore, due to the way we typically express natural language, it is unlikely that the word "mammal" would be learned as a distribution that encompasses both "cat" and "dog", since "mammal" rarely occurs in similar contexts.

Rather than relying on the information from word occurrences, one can do *supervised* training of density embeddings on hierarchical data. In this paper, we propose new training methodology to enable effective supervised probabilistic embeddings. Despite providing rich and intuitive word representations, with a natural ability to represent order relationships, probabilistic embeddings have only been considered in a small number of pioneering works such as Vilnis & McCallum (2014), and these works are almost exclusively focused on *unsupervised embeddings*. Probabilistic Gaussian embeddings trained directly on labeled data have been briefly considered but perform surprisingly poorly compared to other competing models (Vendrov et al., 2015; Vulić et al., 2016).

Our work reaches a very different conclusion: probabilistic Gaussian embeddings can be *highly effective* at capturing ordering and are suitable for modeling hierarchical structures, and can even achieve state-of-the-art results on hypernym prediction and graded lexical entailment tasks, so long as one uses the right training procedures.

In particular, we make the following contributions.

(a) We adopt a new form of loss function for training hierarchical probabilistic order embeddings.

(b) We introduce the notion of soft probabilistic encapsulation orders and a thresholded divergence-based penalty function, which do not over-penalize words with a sufficient encapsulation.

(c) We introduce a new graph-based scheme to select negative samples to contrast the true relationship pairs during training. This approach incorporates hierarchy information to the negative samples that help facilitate training and has added benefits over the hierarchy-agnostic sampling schemes previously used in literature.

(d) We also demonstrate that initializing the right variance scale is highly important for modeling hierarchical data via distributions, allowing the model to exhibit meaningful encapsulation orders.

The outline of our paper is as follows. In Section 2, we introduce the background for Gaussian embeddings (Vilnis & McCallum, 2014) and vector order embeddings (Vendrov et al., 2015). We describe our training methodology in Section 3, where we introduce the notion of soft encapsulation orders (Section 3.2) and explore different divergence measures such as the expected likelihood kernel, KL divergence, and a family of Rényi alpha divergences (Section 3.3). We describe the experiment details in Section 4 and offer a qualitative evaluation of the model in Section 4.3, where we show the visualization of the density encapsulation behavior. We show quantitative results on the WORDNET Hypernym prediction task in Section 4.2 and a graded entailment dataset HYPERLEX in Section 4.4.

In addition, we conduct experiments to show that our proposed changes to learn Gaussian embeddings contribute to the increased performance. We demonstrate (a) the effects of our loss function in Section A.2.3, (b) soft encapsulation in Section A.2.1, (c) negative sample selection in Section 4.4], and (d) initial variance scale in Section A.2.2.

We make our code publicly available.[1]

## 2   BACKGROUND AND RELATED WORK

### 2.1   GAUSSIAN EMBEDDINGS

Vilnis & McCallum (2014) was the first to propose using probability densities as word embeddings. In particular, each word is modeled as a Gaussian distribution, where the mean vector represents the semantics and the covariance describes the uncertainty or nuances in the meanings. These embeddings are trained on a natural text corpus by maximizing the similarity between words that are in the same

---

[1]https://github.com/benathi/density-order-emb

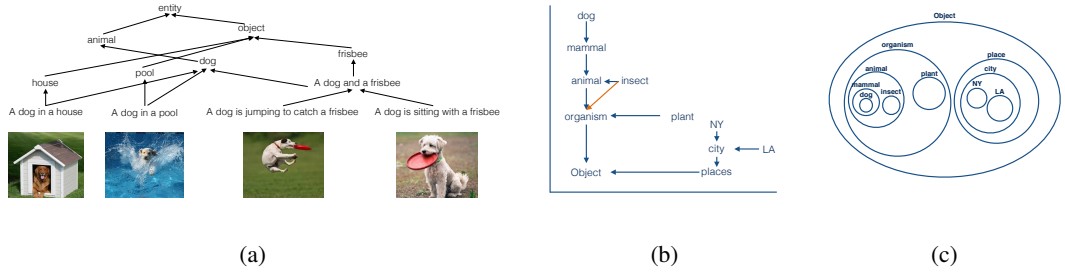

(a)                                    (b)                                    (c)

Figure 1: (a) Captions and images in the visual-semantic hierarchy. (b) Vector order embedding (Vendrov et al., 2015) where specific entities have higher coordinate values. (c) Density order embedding where specific entities correspond to concentrated distributions encapsulated in broader distributions of general entities.

local context of sentences. Given a word $w$ with a true context word $c_p$ and a randomly sampled word $c_n$ (negative context), Gaussian embeddings are learned by minimizing the rank objective in Equation 1, which pushes the similarity of the true context pair $E(w, c_p)$ above that of the negative context pair $E(w, c_n)$ by a margin $m$.

$$L_m(w, c_p, c_n) = \max(0, m - E(w, c_p) + E(w, c_n)) \tag{1}$$

The similarity score $E(u, v)$ for words $u, v$ can be either $E(u, v) = -\text{KL}(f_u, f_v)$ or $E(u, v) = \log\langle f_u, f_u\rangle_{L_2}$ where $f_u, f_v$ are the distributions of words $u$ and $v$, respectively. The Gaussian word embeddings contain rich semantic information and performs competitively in many word similarity benchmarks.

The true context word pairs $(w, c_p)$ are obtained from natural sentences in a text corpus such as Wikipedia. In some cases, specific words can be replaced by a general word in a similar context. For instance, "I love cats" or "I love dogs" can be replaced with "I love animals". Therefore, the trained word embeddings exhibit lexical entailment patterns where specific words such as "dog" and "cat" are concentrated distributions that are encompassed by a more dispersed distribution of "animal", a word that "cat" and "dog" entail. The broad distribution of a general word agrees with the *distributional informativeness hypothesis* proposed by Santus et al. (2014), which says that a generic word can occur in more general contexts in place of the specific ones that entail it.

However, some word entailment pairs have weak density encapsulation patterns due to the nature of word diction. For instance, even though "dog" and "cat" both entail "mammal", it is rarely the case that we observe a sentence "I have a mammal" as opposed to "I have a cat" in a natural corpus; therefore, after training density word embeddings on word occurrences, encapsulation of some true entailment instances do not occur.

## 2.2 PARTIAL ORDERS AND VECTOR ORDER EMBEDDINGS

We describe partial order and the concept of order embeddings proposed by Vendrov et al. (2015), which is highly related to our model.

A partial order over a set of points $X$ is a binary relation $\preceq$ such that for $a, b, c \in X$, the following properties hold: (1) $a \preceq a$ (reflexivity); (2) if $a \preceq b$ and $b \preceq a$ then $a = b$ (antisymmetry); and (3) if $a \preceq b$ and $b \preceq c$ then $a \preceq c$ (transitivity). An example of a partially ordered set is a set of nodes in a tree where $a \preceq b$ means $a$ is a child node of $b$. This concept has applications in natural data such as lexical entailment. For words $a$ and $b$, $a \preceq b$ means that every instance of $a$ is an instance of $b$, or we can say that $a$ entails $b$. We also say that $(a, b)$ has a *hypernym* relationship where $a$ is a hyponym of $b$ and $b$ is a hypernym of $a$. This relationship is asymmetric since $a \preceq b$ does not necessarily imply $(b \preceq a)$. For instance, `aircraft` $\preceq$ `vehicle` but it is not true that `vehicle` $\preceq$ `aircraft`.

An order-embedding is a function $f : (X, \preceq_X) \to (Y, \preceq_Y)$ where $a \preceq_X b$ if and only if $f(a) \preceq_Y f(b)$. Vendrov et al. (2015) proposes to learn the embedding $f$ on $Y = \mathbb{R}_+^N$ where all coordinates are non-negative. Under $\mathbb{R}_+^N$, there exists a partial order relation called the *reversed product order on $\mathbb{R}_+^N$*: $x \preceq y$ if and only if $\forall i, x_i \geq y_i$. That is, a point $x$ entails $y$ if and only if all the coordinate values of $x$ is higher than $y$'s. The origin represents the most general entity at the top of the order hierarchy and

the points further away from the origin become more specific. Figure 1b demonstrates the vector order embeddings on $\mathbb{R}_+^N$. We can see that since `insect` $\preceq$ `animal` and `animal` $\preceq$ `organism`, we can infer directly from the embedding that `insect` $\preceq$ `organism` (orange line, diagonal line). To learn the embeddings, Vendrov et al. (2015) proposes a penalty function $E(x, y) = || \max(0, y - x)||^2$ for a pair $x \preceq y$ which has the property that it is positive if and only if the order is violated.

## 2.3 OTHER RELATED WORK

Li et al. (2017) extends Vendrov et al. (2015) for knowledge representation on data such as ConceptNet (Speer et al., 2016). Another related work by Hockenmaier & Lai (2017) embeds words and phrases in a vector space and uses denotational probabilities for textual entailment tasks. Our models offer an improvement on order embeddings and can be applicable to such tasks, which we aim to explore in future work.

## 3 METHODOLOGY

In Section 3.1, we describe the partial orders that can be induced by density encapsulation. Section 3.2 describes our training approach that softens the notion of strict encapsulation with a viable penalty function.

## 3.1 STRICT ENCAPSULATION PARTIAL ORDERS

A partial order on probability densities can be obtained by the notion of encapsulation. That is, a density $f$ is more specific than a density $g$ if $f$ is encompassed in $g$. The degree of encapsulation can vary, which gives rise to multiple order relations. We define an order relation $\preceq_\eta$ for $\eta \geq 0$ where $\eta$ indicates the degree of encapsulation required for one distribution to entail another. More precisely, for distributions $f$ and $g$,

$$f \preceq_\eta g \Leftrightarrow \{x : f(x) > \eta\} \subseteq \{x : g(x) > \eta\}. \tag{2}$$

Note that $\{x : f(x) > \eta\}$ is a set where the density $f$ is greater than the threshold $\eta$. The relation in Equation 2 says that $f$ entails $g$ if and only if the set of $g$ contains that of $f$. In Figure 2, we depict two Gaussian distributions with different mean vectors and covariance matrices. Figure 2 (left) shows the density values of distributions $f$ (narrow, blue) and $g$ (broad, orange) and different threshold levels. Figure 2 (right) shows that different $\eta$'s give rise to different partial orders. For instance, we observe that neither $f \preceq_{\eta_1} g$ nor $g \preceq_{\eta_1} f$ but $f \preceq_{\eta_3} g$.

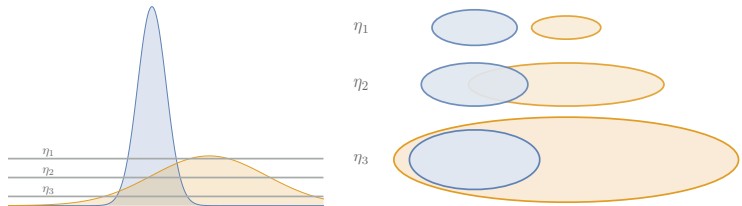

Figure 2: Strict encapsulation orders induced by different $\eta$ values.

## 3.2 SOFT ENCAPSULATION ORDERS

A plausible penalty function for the order relation $f \preceq_\eta g$ is a set measure on $\{x : f(x) > \eta\} - \{x : g(x) > \eta\}$. However, this is difficult to calculate for most distributions, including Gaussians. Instead, we use simple penalty functions based on asymmetric divergence measures between probability densities. Divergence measures $D(\cdot||\cdot)$ have a property that $D(f||g) = 0$ if and only if $f = g$. Using $D(\cdot||\cdot)$ to represent order violation is undesirable since the penalty should be 0 if $f \neq g$ but $f \preceq g$. Therefore, we propose using a thresholded divergence

$$d_\gamma(f, g) = \max(0, D(f||g) - \gamma),$$

which can be zero if $f$ is properly encapsulated in $g$. We discuss the effectiveness of using divergence thresholds in Section A.2.1.

We note that by using $d_\gamma(\cdot, \cdot)$ as a violation penalty, we no longer have the strict *partial order*. In particular, the notion of transitivity in a partial order is not guaranteed. For instance, if $f \preceq g$ and $g \preceq h$, our density order embeddings would yield $d_\gamma(f, g) = 0$ and $d_\gamma(g, h) = 0$. However, it is not necessarily the case that $d_\gamma(f, h) = 0$ since $D(f||h)$ can be greater than $\gamma$. This is not a drawback since a high value of $D(f||h)$ reflects that the hypernym relationship is not direct, requiring many edges from $f$ to $h$ in the hierarchy. The extent of encapsulation contains useful entailment information, as demonstrated in Section 4.4 where our model scores highly correlate with the annotated scores of a challenging lexical entailment dataset and achieves state-of-the-art result.

Another property, antisymmetry, does not strictly hold since if $d_\gamma(f, g) = 0$ and $d_\gamma(g, f) = 0$ does not imply $f = g$. However, in this situation, it is necessary that $f$ and $g$ overlap significantly if $\gamma$ is small. Due to the fact that the $d_\gamma(\cdot, \cdot)$ does not strictly induce a partial order, we refer to this model as *soft density order embeddings* or simply *density order embeddings*.

### 3.3 DIVERGENCE MEASURES

#### 3.3.1 ASYMMETRIC DIVERGENCE

**Kullback-Leibler (KL) Divergence** The KL divergence is an asymmetric measure of the difference between probability distributions. For distributions $f$ and $g$, $\mathrm{KL}(g||f) \equiv \int g(x) \log \frac{g(x)}{f(x)} \, dx$ imposes a high penalty when there is a region of points $x$ such that the density $f(x)$ is low but $g(x)$ is high. An example of such a region is the area on the left of $f$ in Figure 2. This measure penalizes the situation where $f$ is a concentrated distribution relative to $g$; that is, if the distribution $f$ is encompassed by $g$, then the KL yields high penalty. For $d$-dimensional Gaussians $f = \mathcal{N}_d(\mu_f, \Sigma_f)$ and $g = \mathcal{N}_d(\mu_g, \Sigma_g)$,

$$2D_{KL}(f||g) = \log(\det(\Sigma_g)/\det(\Sigma_f)) - d + \mathrm{tr}(\Sigma_g^{-1}\Sigma_f) + (\mu_f - \mu_g)^T \Sigma_g^{-1}(\mu_f - \mu_g) \quad (3)$$

**Rényi $\alpha$-Divergence** is a general family of divergence with varying scale of zero-forcing penalty (Renyi, 1961). Equation 4 describes the general form of the $\alpha$-divergence for $\alpha \neq 0, 1$ (Liese & Vajda, 1987). We note that for $\alpha \to 0$ or 1, we recover the KL divergence and the reverse KL divergence; that is, $\lim_{\alpha \to 1} D_\alpha(f||g) = \mathrm{KL}(f||g)$ and $\lim_{\alpha \to 0} D_\alpha(f||g) = \mathrm{KL}(g||f)$ (Pardo, 2006). The $\alpha$-divergences are asymmetric for all $\alpha$'s, except for $\alpha = 1/2$.

$$D_\alpha(f||g) = \frac{1}{\alpha(\alpha - 1)} \log \left( \int \frac{f(x)^\alpha}{g(x)^{\alpha - 1}} \, dx \right) \quad (4)$$

For two multivariate Gaussians $f$ and $g$, we can write the Rényi divergence as (Pardo, 2006):

$$2D_\alpha(f||g) = -\frac{1}{\alpha(\alpha - 1)} \log \frac{\det(\alpha\Sigma_g + (1 - \alpha)\Sigma_f)}{\left(\det(\Sigma_f)^{1-\alpha} \cdot \det(\Sigma_g)^\alpha\right)} + (\mu_f - \mu_g)^T (\alpha\Sigma_g + (1-\alpha)\Sigma_f)^{-1}(\mu_f - \mu_g).$$
$$(5)$$

The parameter $\alpha$ controls the degree of *zero forcing* where minimizing $D_\alpha(f||g)$ for high $\alpha$ results in $f$ being more concentrated to the region of $g$ with high density. For low $\alpha$, $f$ tends to be *mass-covering*, encompassing regions of $g$ including the low density regions. Recent work by Li & Turner (2016) demonstrates that different applications can require different degrees of zero-forcing penalty.

#### 3.3.2 SYMMETRIC DIVERGENCE

**Expected Likelihood Kernel** The expected likelihood kernel (ELK) (Jebara et al., 2004) is a symmetric measure of affinity, define as $K(f, g) = \langle f, g \rangle_\mathcal{H}$. For two Gaussians $f$ and $g$,

$$2 \log \langle f, g \rangle_\mathcal{H} = -\log \det(\Sigma_f + \Sigma_g) - d \log(2\pi) - (\mu_f - \mu_g)^T (\Sigma_f + \Sigma_g)^{-1}(\mu_f - \mu_g) \quad (6)$$

Since this kernel is a similarity score, we use its negative as our penalty. That is, $D_{\mathrm{ELK}}(f||g) = -2 \log \langle f, g \rangle_\mathcal{H}$. Intuitively, the asymmetric measures should be more successful at training density order embeddings. However, a symmetric measure can result in the encapsulation order as well since a general entity often has to minimize the penalty with many specific elements and consequently ends up having a broad distribution to lower the average loss. The expected likelihood kernel is used to train Gaussian and Gaussian Mixture word embeddings on a large text corpus (Vilnis & McCallum, 2014; Athiwaratkun & Wilson, 2017) where the model performs well on the word entailment dataset (Baroni et al., 2012).

## 3.4 LOSS FUNCTION

To learn our density embeddings, we use a loss function similar to that of Vendrov et al. (2015). Minimizing this function (Equation 7) is equivalent to minimizing the penalty between a true relationship pair $(u, v)$ where $u \preceq v$, but pushing the penalty to be above a margin $m$ for the negative example $(u', v')$ where $u' \not\preceq v'$:

$$\sum_{(u,v)\in\mathcal{D}} d(u, v) + \max\{0, m - d(u', v')\} \tag{7}$$

We note that this loss function is different than the rank-margin loss introduced in the original Gaussian embeddings (Equation 1). Equation 7 aims to reduce the dissimilarity of a true relationship pair $d(u, v)$ with no constraint, unlike in Equation 1, which becomes zero if $d(u, v)$ is above $d(u', v')$ by margin $m$.

## 3.5 SELECTING NEGATIVE SAMPLES

In many embedding models such as WORD2VEC (Mikolov et al., 2013) or Gaussian embeddings (Vilnis & McCallum, 2014), negative samples are often used in the training procedure to contrast with true samples from the dataset. For flat data such as words in a text corpus, negative samples are selected randomly from a unigram distribution. We propose new graph-based methods to select negative samples that are suitable for hierarchical data, as demonstrated by the improved performance of our density embeddings. In our experiments, we use various combinations of the following methods.

Method **S1**: A simple negative sampling procedure used by Vendrov et al. (2015) is to replace a true hypernym pair $(u, v)$ with either $(u, v')$ or $(u', v)$ where $u', v'$ are randomly sampled from a uniform distribution of vertices. Method **S2**: We use a negative sample $(v, u)$ if $(u, v)$ is a true relationship pair. The motivation is due to the fact that it is important to make $D(v||u)$ higher than $D(u||v)$ in order to distinguish the directionality of density encapsulation. Method **S3**: It is important to increase the divergence between neighbor entities that do not entail each other. Let $A(w)$ denote all descendants of $w$ in the training set $\mathcal{D}$, including $w$ itself. We first randomly sample an entity $w \in \mathcal{D}$ that has at least 2 descendants and randomly select a descendant $u \in A(w) - \{w\}$. Then, we randomly select an entity $v \in A(w) - A(u)$ and use the random neighbor pair $(v, u)$ as a negative sample. Note that we can have $u \preceq v$, in which case the pair $(v, u)$ is a reverse relationship. Method **S4**: Same as **S3** except that we sample $v \in A(w) - A(u) - \{w\}$, which excludes the possibility of drawing $(w, u)$.

## 4 EXPERIMENTS

We have introduced density order embeddings to model hierarchical data via encapsulation of probability densities. We propose using a new loss function, graph-based negative sample selections, and a penalty relaxation to induce soft partial orders. In this section, we show the effectiveness of our model on WORDNET hypernym prediction and a challenging graded lexical entailment task, where we achieve state-of-the-art performance.

First, we provide the training details in Section 4.1 and describe the hypernym prediction experiment in 4.2. We offers insights into our model with the qualitative analysis and visualization in Section 4.3. We evaluate our model on HYPERLEX, a lexical entailment dataset in Section 4.4.

## 4.1 TRAINING DETAILS

We have a similar data setup to the experiment by Vendrov et al. (2015) where we use the transitive closure of WORDNET noun hypernym relationships which contains $82,115$ synsets and $837,888$ hypernym pairs from $84,427$ direct hypernym edges. We obtain the data using the WORDNET API of NLTK version 3.2.1 (Loper & Bird, 2002).

The validation set contains $4000$ true hypernym relationships as well as $4000$ false hypernym relationships where the false hypernym relationships are constructed from the **S1** negative sampling

described in Section 3.5. The same process applies for the test set with another set of $4000$ true hypernym relationships and $4000$ false hypernym relationships.

We use $d$-dimensional Gaussian distributions with diagonal covariance matrices. We use $d = 50$ as the default dimension and analyze the results using different $d$'s in Section A.2.4. We initialize the mean vectors to have a unit norm and normalize the mean vectors in the training graph. We initialize the diagonal variance components to be all equal to $\beta$ and optimize on the unconstrained space of $\log(\Sigma)$. We discuss the important effects of the initial variance scale in Section A.2.2.

We use a minibatch size of $500$ true hypernym pairs and use varying number of negative hypernym pairs, depending on the negative sample combination proposed in Section 3.5. We discuss the results for many selection strategies in Section 4.4. We also experiment with multiple divergence measures $D(\cdot||\cdot)$ described in Section 3.3. We use $D(\cdot||\cdot) = D_{KL}(\cdot||\cdot)$ unless stated otherwise. Section A.2.5 discuss the results using the $\alpha$-divergence family with varying degrees of zero-forcing parameter $\alpha$'s. We use the Adam optimizer (Kingma & Ba, 2014) and train our model for at most $20$ epochs. For each energy function, we tune the hyperparameters on grids. The hyperparameters are the loss margin $m$, the initial variance scale $\beta$, and the energy threshold $\gamma$. We evaluate the results by computing the penalty on the validation set to find the best threshold for binary classification, and use this threshold to perform prediction on the test set. Section A.1 describes the hyperparameters for all our models.

## 4.2 HYPERNYM PREDICTION

We show the prediction accuracy results on the test set of WORDNET hypernyms in Table 1. We compare our results with **vector order-embeddings** (VOE) by Vendrov et al. (2015) where the model details are explained in Section 2.2. Another important baseline is the **transitive closure**, which requires no learning and classifies if a held-out edge is a hypernym relationship by determining if it is in the union of the training edges. **word2gauss** and **word2gauss**† are the Gaussian embeddings trained using the loss function in Vilnis & McCallum (2014) (Equation 1) where **word2gauss** is the result reported by Vendrov et al. (2015) and **word2gauss**† is the best performance of our replication (see Section A.2.3 for more details). Our density order embedding (DOE) outperforms the implementation by Vilnis & McCallum (2014) significantly; this highlights the fact that a different approach to train Gaussian representations might be required for a different task.

We observe that the symmetric model (ELK) performs quite well for this task despite the fact that the symmetric metric cannot capture directionality. In particular, ELK can detect pairs of concepts with no relationships well when they're far away in the density space. In addition, for pairs that are related, ELK can detect pairs that overlap significantly in density space. The lack of directionality has more pronounced effects in the graded lexical entailment task (Section 4.4) where we observe a high degradation in performance if ELK is used instead of KL.

Our method also outperforms the vector order embeddings (VOE). We also include the results for a 2-dimensional Gaussian embedding trained for the purpose of visualization (Section 4.3). Surprisingly, the performance is very strong, beating the transitive closure and other baselines except VOE while only having 4 parameters: 2 from 2-dimensional $\mu$ and another 2 from the diagonal $\Sigma$. The results using a symmetric measure also outperforms the baselines but has a slightly lower accuracy than the asymmetric model.

Figure 3 offers an explanation as to why our density order embeddings might be easier to learn, compared to the vector counterpart. In certain cases such as fitting a general concept `entity` to the embedding space, we simply need to adjust the distribution of `entity` to be broad enough to encompass all other concepts. In the vector counterpart, it might be required to shift many points further from the origin to accommodate `entity` to reduce cascading order violations.

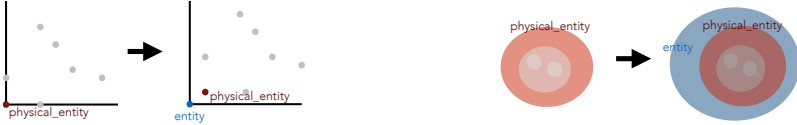

Figure 3: (**Left**) Adding a concept `entity` to vector order embedding (**Right**) Adding a concept `entity` to density order embedding

Table 1: Classification accuracy on hypernym relationship test set from WordNet.

| Method | Test Accuracy (%) |
| --- | --- |
| transitive closure | 88.2 |
| word2gauss | 86.6 |
| word2gauss† | 88.6 |
| VOE (symmetric) | 84.2 |
| VOE | 90.6 |
| DOE (ELK) | 92.1 |
| DOE (KL, reversed) | 83.2 |
| DOE (KL) | **92.3** |
| DOE (KL, $d = 2$) | 89.2 |

## 4.3 QUALITATIVE ANALYSIS

For qualitative analysis, we additionally train a 2-dimensional Gaussian model for visualization. Our qualitative analysis shows that the encapsulation behavior can be observed in the trained model. Figure 4 demonstrates the ordering of synsets in the density space. Each ellipse represents a Gaussian distribution where the center is given by the mean vector $\mu$ and the major/minor axes are given by the diagonal standard deviations $\sqrt{\Sigma}$, scaled by 300 and 30 for $x$ and $y$ axis for visibility.

Most hypernym relationships exhibit the encapsulation behavior where the hypernym encompasses the synset that entails it. For instance, the distribution of `whole.n.02` is subsumed in the distribution of `physical_entity.n.01`. Note that `location.n.01` is not entirely encapsulated by `physical_entity.n.01` under this visualization. However, we can still predict which entity should be the hypernym among the two since the KL divergence of one given another would be drastically different. This is because a large part of `physical_entity.n.01` has considerable density at the locations where location.n.01 has very low density. This causes KL(`physical_entity.n.01` ‖ `location.n.01`) to be very high (5103) relative to KL(`location.n.01` ‖ `physical_entity.n.01`) (206). Table 2 shows the KL values for all pairs where we note that the numbers are from the full model ($d = 50$).

Another interesting pair is `city.n.01` $\preceq$ `location.n.01` where we see the two distributions have very similar contours and the encapsulation is not as distinct. In our full model $d = 50$, the distribution of `location.n.01` encompasses `city.n.01`'s, indicated by low KL(`city.n.01`‖`location.n.01`) but high KL(`location.n.01`‖`city.n.01`).

Figure 4 (**Right**) demonstrates the idea that synsets on the top of the hypernym hierarchy usually have higher "volume". A convenient metric that reflects this quantity is $\log \det(\Sigma)$ for a Gaussian distribution with covariance $\Sigma$. We can see that the synset, `physical_entity.n.01`, being the hypernym of all the synsets shown, has the highest $\log \det(\Sigma)$ whereas entities that are more specific such as `object.n.01`, `whole.n.02` and `living_thing` have decreasingly lower volume.

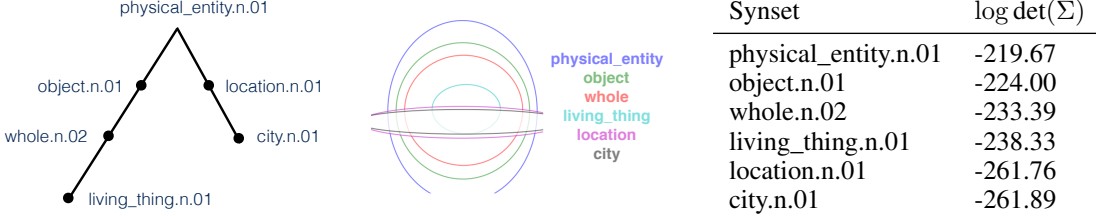

| Synset | $\log \det(\Sigma)$ |
| --- | --- |
| physical_entity.n.01 | -219.67 |
| object.n.01 | -224.00 |
| whole.n.02 | -233.39 |
| living_thing.n.01 | -238.33 |
| location.n.01 | -261.76 |
| city.n.01 | -261.89 |

Figure 4: [best viewed electronically] (**Left**) Synsets and their hypernym relationships from WordNet. (**Middle**) Visualization of our 2-dimensional Gaussian order embedding. (**Right**) The Gaussian "volume" ($\log \det \Sigma$) of the 50-dimensional Gaussian model.

Table 2: KL(column‖row). Cells in boldface indicate true WORDNET hypernym relationships (column $\preceq$ row). Our model predicts a synset pair as a hypernym if the KL less than 1900, where this value is tuned based on the validation set. Most relationship pairs are correctly predicted except for the underlined cells.

| | city | location | living_thing | whole | object | physical_entity |
|---|---|---|---|---|---|---|
| city | 0 | 1025 | 4999 | 4673 | 23673 | 4639 |
| location | **159** | 0 | 4324 | 4122 | 26121 | 5103 |
| living_thing | 3623 | 6798 | 0 | 1452 | 2953 | 5936 |
| whole | 3033 | 6367 | **66** | 0 | 6439 | 6682 |
| object | 138 | 80 | **125** | **77** | 0 | 6618 |
| physical_entity | **232** | **206** | **193** | **166** | **152** | 0 |

## 4.4 GRADED LEXICAL ENTAILMENT

HYPERLEX is a lexical entailment dataset which has fine-grained human annotated scores between concept pairs, capturing varying degrees of entailment (Vulić et al., 2016). Concept pairs in HYPERLEX reflect many variants of hypernym relationships, such as no-rel (no lexical relationship), ant (antonyms), syn (synonyms), cohyp (sharing a hypernym but not a hypernym of each other), hyp (hypernym), rhyp (reverse hypernym). We use the noun dataset of HYPERLEX for evaluation, which contains 2,163 pairs.

We evaluate our model by comparing our model scores against the annotated scores. Obtaining a high correlation on a fine-grained annotated dataset is a much harder task compared to a binary prediction since performing well requires meaningful model scores in order to reflect nuances in hypernymy. We use negative divergence as our score for hypernymy scale where large values indicate high degrees of entailment.

We note that the concepts in our trained models are WORDNET synsets, where each synset corresponds to a specific meaning of a word. For instance, pop.n.03 has a definition "a sharp explosive sound as from a gunshot or drawing a cork" whereas pop.n.04 corresponds to "music of general appeal to teenagers; ...". For a given pair of words $(u, v)$, we use the score of the synset pair $(s'_u, s'_v)$ that has the lowest KL divergence among all the pairs $S_n \times S_v$ where $S_u, S_v$ are sets of synsets for words $u$ and $v$, respectively. More precisely, $s(u, v) = -\min_{s_u \in S_u, s_v \in S_v} D(s_u, s_v)$. This pair selection corresponds to choosing the synset pair that has the highest degree of entailment. This approach has been used in word embeddings literature to select most related word pairs (Athiwaratkun & Wilson, 2017). For word pairs that are not in the model, we assign the score equal to the median of all scores. We evaluate our model scores against the human annotated scores using Spearman's rank correlation.

Table 3 shows HYPERLEX results of our models **DOE-A** (asymmetric) and **DOE-S** (symmetric) as well as other competing models. The model **DOE-A** which uses KL divergence and negative sampling approach **S1**, **S2** and **S4** outperforms all other existing models, achieving state-of-the-art performance for the HYPERLEX noun dataset. (See Section A.1 for hyperparameter details) The model **DOE-S** which uses expected likelihood kernel attains a lower score of 0.455 compared to the asymmetric counterpart (**DOE-A**). This result underscores the importance of asymmetric measures which can capture relationship directionality.

We provide a brief summary of competing models: **FR** scores are based on concept word frequency ratio (Weeds et al., 2004). **SLQS** uses entropy-based measure to quantify entailment (Santus et al., 2014). **Vis-ID** calculates scores based on visual generality measures (Kiela et al., 2015). **WN-B** calculates the scores based on the shortest path between concepts in WN taxonomy (Miller, 1995). **w2g** Guassian embeddings trained using the methodology in Vilnis & McCallum (2014). **VOE** Vector order embeddings (Vendrov et al., 2015). **Euc** and **Poin** calculate scores based on the Euclidean distance and Poincaré distance of the trained Poincaré embeddings (Nickel & Kiela, 2017). The models **FR** and **SLQS** are based on word occurrences in text corpus, where **FR** is trained on the British National Corpus and **SLQS** is trained on UKWAC, WACKYPEDIA (Bailey & Thompson, 2006; Baroni et al., 2009) and annotated BLESS dataset (Baroni & Lenci, 2011). Other models **Vis-ID, w2g, VOE, Euc, Poin** and ours are trained on WordNet, with the exception that **Vis-ID** also uses

Table 3: Spearman's correlation for HYPERLEX nouns.

| | FR | SLQS | Vis-ID | WN-B | w2g | VOE | Poin | HypV | DOE-S | DOE-A |
|---|---|---|---|---|---|---|---|---|---|---|
| $\rho$ | 0.283 | 0.229 | 0.253 | 0.240 | 0.192 | 0.195 | 0.512 | 0.540 | 0.455 | **0.590** |

Table 4: Spearman's correlation for HYPERLEX nouns for different negative sample schemes.

| Negative Samples | $\rho$ | Negative Samples | $\rho$ |
|---|---|---|---|
| $1\times$**S1** | 0.527 | $1\times$**S1** + **S2** + **S4** | **0.590** |
| $2\times$**S1** | 0.529 | $2\times$**S1** + **S2** + **S4** | 0.580 |
| $5\times$**S1** | 0.518 | $5\times$**S1** + **S2** + **S4** | 0.582 |
| $10\times$**S1** | 0.517 | $1\times$**S1** + **S2** + **S3** | 0.570 |
| $1\times$**S1** + **S2** | 0.567 | $2\times$**S1** + **S2** + **S3** | 0.581 |
| $2\times$**S1** + **S2** | 0.567 | **S1** + $0.1\times$**S2** + $0.9\times$**S3** | 0.564 |
| $3\times$**S1** + **S2** | 0.584 | **S1** + $0.3\times$**S2** + $0.7\times$**S3** | 0.574 |
| $5\times$**S1** + **S2** | 0.561 | **S1** + $0.7\times$**S2** + $0.3\times$**S3** | 0.555 |
| $10\times$**S1** + **S2** | 0.550 | **S1** + $0.9\times$**S2** + $0.1\times$**S3** | 0.533 |

Google image search results for visual data. The reported results of **FR**, **SLQS**, **Vis-ID**, **WN-B**, **w2g** and **VOE** are from Vulić et al. (2016).

We note that an implementation of Gaussian embeddings model (**w2g**) reported by Vulić et al. (2016) does not perform well compared to previous benchmarks such as **Vis-ID, FR, SLQS**. Our training approach yields the opposite results and outperforms other highly competitive methods such as Poincaré embeddings and Hypervec. This underscores the fact that training approach matters a great deal, even if the concept representation of our work and Vilnis & McCallum (2014)'s are both Gaussian distributions. In addition, we also observe that the vector order embeddings (VOE) do not perform well compared to our model. We hypothesize that it is due to the "soft" orders induced by the divergence penalty that allows our model scores to reflect more closely with hypernymy degrees.

We note another interesting observation that a model trained on a symmetric divergence (ELK) from Section 4.2 can also achieve a high HYPERLEX correlation of $0.532$ if KL is used to calculate the model scores. This is because the encapsulation behavior can arise even though the training penalty is symmetric (more explanation in Section 4.2). However, using the symmetric divergence based on ELK results in poor performance on HYPERLEX (0.455), which is expected since it cannot capture the directionality of hypernymy.

We note that another model LEAR obtains an impressive score of $0.686$ (Vulić & Mrkšić, 2014). However, LEAR use pre-trained word embeddings such as WORD2VEC or GLOVE as a pre-processing step, leveraging a large vocabulary with rich semantic information. To the best of our knowledge, our model achieves the highest HYPERLEX Spearman's correlation among models without using large-scale pre-trained embeddings.

Table 4 shows the effects of negative sample selection described in Section 3.5. We note again that **S1** is the technique used in literature Socher et al. (2013); Vendrov et al. (2015) and **S2**, **S3**, **S4** are the new techniques we proposed. The notation, for instance, $k \times$ **S1** + **S2** corresponds to using $k$ samples from **S1** and 1 sample from **S2** per each positive sample. We observe that our new selection methods offer strong improvement from the range of $0.51 - 0.52$ (using **S1** alone) to $0.55$ or above for most combinations with our new selection schemes.

## 5 FUTURE WORK

Analogous to recent work by Vulić & Mrkšić (2014) which post-processed word embeddings such as GLOVE or WORD2VEC, our future work includes using the WordNet hierarchy to impose encapsulation orders when training probabilistic embeddings.

In the future, the distribution approach could also be developed for encoder-decoder based models for tasks such as caption generation where the encoder represents the data as a distribution, containing semantic and visual features with uncertainty, and passes this distribution to the decoder which maps to text or images. Such approaches would be reminiscent of variational autoencoders (Kingma & Welling, 2013), which take *samples* from the encoder's distribution.

ACKNOWLEDGEMENTS

We thank NSF IIS-1563887 for support.

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

# A    Supplementary Materials

## A.1    Model Hyperparameters

In Section 4.3, the $2-$dimensional Gaussian model is trained with **S-1** method where the number of negative samples is equal to the number of positive samples. The best hyperparameters for $d = 2$ model is $(m, \beta, \gamma) = (100.0, 2 \times 10^{-4}, 3.0)$.

In Section 4.2, the best hyperparameters $(m, \beta, \gamma)$ for each of our model are as follows: For Gaussian with KL penalty: $(2000.0, 5 \times 10^{-5}, 500.0)$, , Gaussian with reversed KL penalty: $(1000.0, 1 \times 10^{-4}, 1000.0)$, Gaussian with ELK penalty $(1000, 1 \times 10^{-5}, 10)$.

In Section 4.4, we use the same hyperparameters as in 4.2 with KL penalty, but a different negative sample combination in order to increase the distinguishability of divergence scores. For each positive sample in the training set, we use one sample from each of the methods **S1**, **S2**, **S3**. We note that the model from Section 4.2, using **S1** with the KL penalty obtains a Spearman's correlation of 0.527.

## A.2    Analysis of Training Methodology

We emphasize that Gaussian embeddings have been used in the literature, both in the unsupervised settings where word embeddings are trained with local contexts from text corpus, and in supervised settings where concept embeddings are trained to model annotated data such as WORDNET . The results in supervised settings such as modeling WORDNET have been reported to compare with competing models but often have inferior performance (Vendrov et al., 2015; Vulić et al., 2016). Our paper reaches the opposite conclusion, showing that a different training approach using Gaussian representations can achieve state-of-the-art results.

### A.2.1    Divergence Threshold

Consider a relationship $f \preceq g$ where $f$ is a hyponym of $g$ or $g$ is a hypernym of $f$. Even though the divergence $D(f||g)$ can capture the extent of encapsulation, a density $f$ will have the lowest divergence with respect with $g$ only if $f = g$. In addition, if $f$ is a more concentrated distribution that is encompassed by $g$, $D(f||g)$ is minimized when $f$ is at the center of $g$. However, if there any many hyponyms $f_1, f_2$ of $g$, the hyponyms can compete to be close to the center, resulting in too much overlapping between $f_1$ and $f_2$ if the random sampling to penalize negative pairs is not sufficiently strong. The divergence threshold $\gamma$ is used such that there is no longer a penalty once the divergence is below a certain level.

We demonstrate empirically that the threshold $\gamma$ is important for learning meaningful Gaussian distributions. We fix the hyperparameters $m = 2000$ and $\beta = 5 \times 10^{-5}$, with **S1** negative sampling. Figure 5 shows that there is an optimal non-zero threshold and yields the best performance for both WORDNET Hypernym prediction and HYPERLEX Spearman's correlation. We observe that using $\gamma = 0$ is detrimental to the performance, especially on HYPERLEX results.

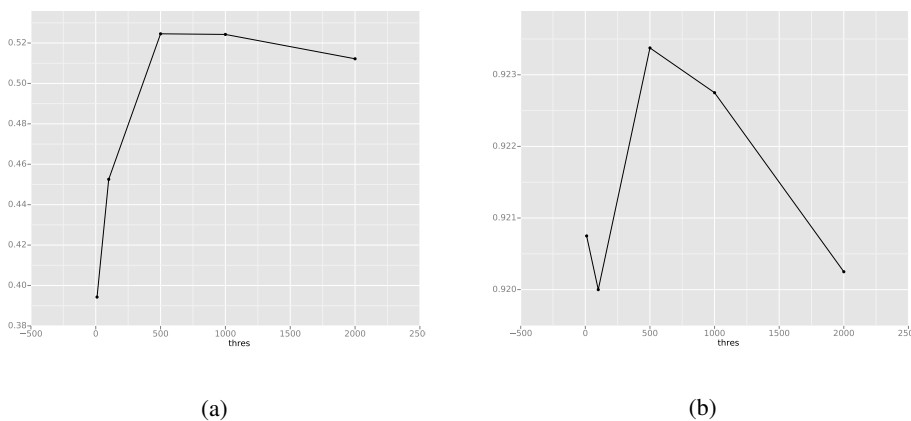

(a)                                                    (b)

Figure 5: (a) Spearman's correlation on HYPERLEX versus $\gamma$ (b) Test Prediction Accuracy versus $\gamma$.

### A.2.2    Initial Variance Scale

As opposed to the mean vectors that are randomly initialized, we initialize all diagonal covariance elements to be the same. Even though the variance can adapt during training, we find that different initial scales of variance

result in drastically different performance. To demonstrate, in Figure 6, we show the best test accuracy and HYPERLEX Spearman's correlation for each initial variance scale, with other hyperparameters (margin $m$ and threshold $\gamma$) tuned for each variance. We use **S1 + S2 + S4** as a negative sampling method. In general, a low variance scale $\beta$ increases the scale of the loss and requires higher margin $m$ and threshold $\gamma$. We observe that the best prediction accuracy is obtained when $\log(\beta) \approx -10$ or $\beta = 5 \times 10^{-5}$. The best HYPERLEX results are obtained when the scales of $\beta$ are sufficiently low. The intuition is that low $\beta$ increases the scale of divergence $D(\cdot||\cdot)$, which increases the ability to capture relationship nuances.

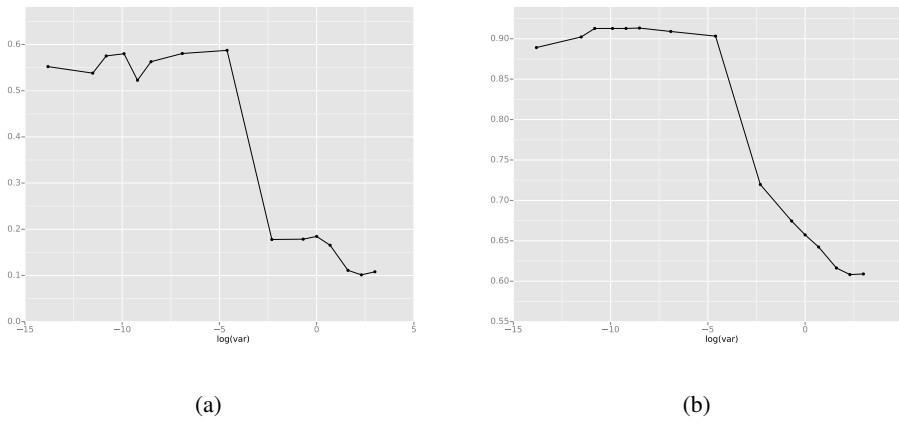

| (a) | (b) |
|---|---|

Figure 6: (a) Spearman's correlation on HYPERLEX versus $\log(\beta)$ (b) Test Prediction Accuracy versus $\log(\beta)$.

### A.2.3 LOSS FUNCTION

We verify that for this task, our loss function in Equation 7 in superior to Equation 1 originally proposed by Vilnis & McCallum (2014). We use the exact same setup with new negative sample selections and KL divergence thresholding and compare the two loss functions. Table 5 verifies our claim.

Table 5: Best results for each loss function for two negative sampling setups: **S1 (Left)** and **S1 + S2 + S4 (Right)**

| | Test Accuracy | HYPERLEX | | | Test Accuracy | HYPERLEX |
|---|---|---|---|---|---|---|
| Eq. 7 | 0.923 | 0.527 | | Eq. 7 | 0.911 | 0.590 |
| Eq. 1 | 0.886 | 0.524 | | Eq. 1 | 0.796 | 0.489 |

### A.2.4 DIMENSIONALITY

Table 6 shows the results for many dimensionalities for two negative sample strategies: **S1** and **S1 + S2 + S4** .

Table 6: Best results for each dimension with negative samples **S1 (Left)** and **S1 + S2 + S4 (Right)**

| $d$ | Test Accuracy | HYPERLEX | | $d$ | Test Accuracy | HYPERLEX |
|---|---|---|---|---|---|---|
| 5 | 0.909 | 0.437 | | 5 | 0.901 | 0.483 |
| 10 | 0.919 | 0.462 | | 10 | 0.909 | 0.526 |
| 20 | 0.922 | 0.487 | | 20 | 0.914 | 0.545 |
| 50 | 0.923 | 0.527 | | 50 | 0.911 | 0.590 |
| 100 | 0.924 | 0.526 | | 100 | 0.913 | 0.573 |
| 200 | 0.918 | 0.526 | | 200 | 0.910 | 0.568 |

### A.2.5 $\alpha$-DIVERGENCES

Table 7 show the results using models trained and evaluated with $D(\cdot||\cdot) = D_\alpha(\cdot||\cdot)$ with negative sampling approach **S1**. Interestingly, we found that $\alpha \to 1$ (KL) offers the best result for both prediction accuracy and HYPERLEX . It is possible that $\alpha = 1$ is sufficiently asymmetric enough to distinguish hypernym directionality, but does not have as sharp penalty as in $\alpha > 1$, which can help learning.

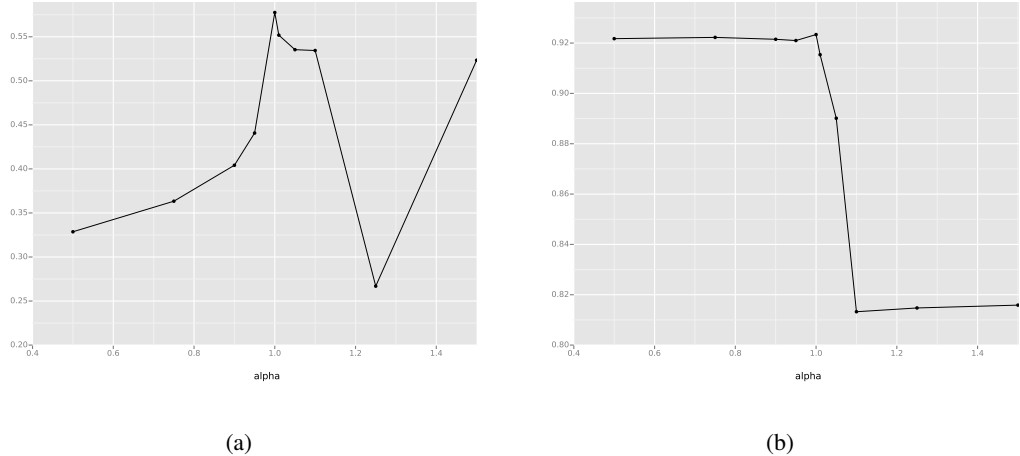

(a)                                             (b)

Figure 7: (a) Spearman's correlation on HYPERLEX versus $\alpha$ (b) Test Prediction Accuracy versus $\alpha$.

