# OpenReview forum: "Hierarchical Density Order Embeddings"
_ICLR.cc/2018/Conference — Accept (Poster)_

### Official Review · AnonReviewer2 · 2017-11-25
**KL is not symmetric but its directional aspect was not found to  be significant for lexical entailment**

**Rating:** 4
**Confidence:** 3

**Review:**

The paper presents a method for hierarchical object embedding by Gaussian densities for lexical entailment tasks.Each word is represented  by a diagonal Gaussian and the KL divergence is used as a directional distance measure. if D(f||g) < gamma then the concept represented by f entails the concept represented by g.

The main technical difference of the present work compared from the  main prior work (Vendrov, 2015) is that in addition to mean vector representation they use here also a variance component. The main modeling challenge here to to define a  good directional measure that can be suitable for lexical entailment. in Vendrov work they defined a partial ordering. Here,  the KL is not symmetric but its directional aspect is not significant.
For example if we set all the variances to be a unit matrix than the KL is collapsed to be a simple symmetrical Euclidean distance. We can also see from Table 1 that if we replace KL by its symmetrical variant we get similar results. Hence, I was not convinced that the propose KL+Gaussian modeling is suitable for directional relations.

The paper also presents several methods for negative samplings and according to table 4 there is a lot of performance variability based on the method that is used for selecting negative sampling. I find this component of the proposed algorithm  very heuristic.

To summarize, I don't think there is enough interesting novelty in this paper. If the focus of the  paper is on  obtaining good entailment results, maybe an NLP conference can be a more suitable venue.

---

> ### Author Response · Authors · 2017-12-30
> **Response to AnonReviewer2**
>
> We thank the reviewer for thoughtful comments.
>
> To address the importance of directionality, we clarify the differences in using symmetric measures to train versus to evaluate.  We also highlight and explain the major empirical differences for the purpose of order embeddings.
>
> [Symmetric Measure for Training] In Section 3.3.2: Symmetric Divergence, we describe why symmetric measures can be used to train embeddings that reflect hierarchy:
>
> “Intuitively, the asymmetric measures should be more successful at training density order embeddings. However, a symmetric measure can result in the encapsulation order as well since a general entity often has to minimize the penalty with many specific elements and consequently ends up having a broad distribution.”
>
> For further explanation, consider a scenario where we have true relationships a -> c and b -> c and use a symmetric training measure to train the embeddings (x -> y denotes x entails y). The desired outcome is such that the distribution of ‘c’ encapsulates the distribution of ‘a’ as well as ‘b’. To satisfy this, the distribution of ‘c’ ends up being broad and encompass both ‘a’ and ‘b’. to lower the average loss.
>
> We emphasize that at evaluation time, if we use symmetric measures to generate entailment scores, we would end up discarding the directionality and thus hurt the performance. This brings us to the next point, where we discuss the empirical differences, underscoring the foundational importance of an asymmetric measure.
>
> [KL for directionality]
>
> The directionality of KL is crucial to capture hierarchical information. In our experiments, when our model is trained with the symmetric ELK distance, using ELK to generate encapsulation scores results in poor performance of 0.455 whereas using KL yields 0.532.  This difference is described in section 4.4.
>
> With regards to the performance of ELK in Table 1, negative log ELK, a symmetric distance metric, can generate meaningful scores for word pairs that do not have directional relationships, such as ‘animal’ and ‘location’. This is because these non-relationship density pairs do not significantly overlap and ELK-based metric would yield high distance value. However, this metric is not suitable for capturing directionality of relationships such as ‘animal’ and ‘dog’.
>
> For example, Table 2 which illustrates the values of KL divergences between many word pairs, underscores the foundational importance of KL’s directionality. For instance, KL(‘object’ | ‘physical_entity’) = 152, whereas the reverse case KL(‘physical_entity’ | ‘object’) = 6618. Based on the prediction threshold of 1900 (selected via validation set), we correctly predict that ‘object’ entails ‘physical_entity’ (since 152 << 1900) but ‘physical_entity’ does not entail ‘object’ (6618 >> 1900). The entailment score function (negative KL) also nicely assigns high scores to the true hypernym pairs such as (‘object’ -> ‘physical_entity’, with score -152) and low scores for the non-hypernym pairs such as (‘physical_entity’ -> ‘object’, with score -6618). This behavior is important to measure the degree of lexical entailment. If the evaluation divergence were symmetric, (‘object’ -> ‘physical_entity’) and (‘physical_entity’ -> ‘object’) would have the same score, which is very undesirable, since these pairs do not have the same degree of entailment.
>
> [unit variance] It is true that if the variance components are all the same (being 1), the KL becomes symmetric. However, our learned distributions tend to have very different covariance matrices.  Figure 4, the log det(Sigma) vary markedly among different concepts. The concepts that are more general such as `physical_entity’ have very high log(det(Sigma)) of -219.67 compared to specific concepts such as ‘city’ with log(det(Sigma)) = -261.89.

---

> > ### Author Response · Authors · 2017-12-30
> > **[Continued] Response to AnonReviewer2**
> >
> > [novelty] Aside from the foundational importance of asymmetry in divergences for probabilistic order embeddings, there is much other interesting novelty in the paper too. We propose new training procedures that learn highly effective Gaussian embeddings in the supervised setting. The new changes include (1) using max-margin loss (Equation 7) instead of rank loss (Equation 1); (2) using a divergence threshold to induce soft partial orders and prevent unnecessary penalization; (3) a new scheme to select negative samples for max-margin loss; (4) investigating other hyperparameters that are important for density encapsulation such as adjusting variance scale; (5) proposing and investigating general alpha divergences as metrics between word densities.  This direction is significant — analogous to exploring Renyi divergences instead of variational KL divergences for approximate Bayesian inference.
> >
> > In general, despite the great promise of probabilistic embeddings, such approaches have not been widely explored, and their use in order embeddings -- where they are perhaps most natural -- is essentially uncharted territory.
> >
> > The empirical results are generally quite strong.  Not all results are positive (for example, we found alpha divergences to not improve on KL), but there is certainly great value in honestly reporting conceptually interesting experiments, especially if there are negative results, which tend to go under-reported.  Many of the results (as discussed above) are also positive.
> >
> > [Effects of new negative sampling] Note that the traditional approach of Vendrov 2015 uses the sampling scheme S1. From Table 4, using S1 alone results in the HyperLex score of at most 0.527. However, using our proposed approaches (S1 together with S2, S3, S4 with different heuristic combination) results in a significant increase in performance in most (if not all) cases, allowing us to achieve a score of 0.590 (~12% increase). We would like to point out that this is a strength of our proposed negative sampling methods (S2, S3, S4) since most combinations provide an increase in performance.

---

### Official Review · AnonReviewer3 · 2017-11-27
**Potentially a good paper, badly presented**

**Rating:** 6
**Confidence:** 4

**Review:**

The paper presents a study on the use of density embedding for modeling hierarchical semantic relations, and in particular on the hypernym one. The goal is to capture hypernyms of some synsets, even if their occurrence is scarce on the training data.
+++pros: 1) potentially a good idea, capable of filling an ontology of relations scarcely present in a given repository 2) solid theoretical background, even if no methodological novelty has been introduced (this is also a cons!)
---cons: 1) Badly presented: the writing of the paper fails in let the reader aware of what the paper actually serves

COMMENTS:
The introduction puzzled me:  the authors, once they stated the problem (the scarceness of the hypernyms' occurrences in the texts w.r.t. their hyponyms), proposed a solution which seems not to directly solve this problem. So I suggest the authors to better explain the connection between the told problem and their proposed solution, and how this can solve the problem.

This aspect is also present in the experiments section, since it is not possible to understand how much the problem (the scarceness of the hypernyms) is present in the HYPERLEX dataset.

How the 4000 hypernyms have been selected? Why a diagonal covariance has been estimated, and not a full covariance one?

n Figure 4 middle, it is not clear whether the location and city concepts are intersecting the other synsets. It shouldn't be, but the authors should spend a little on this.

Apart from these comments, I found the paper interesting especially for the big amount fo comparisons carried out.

As a final general comment, I would have appreciated a paper more self explanative, without referring to the paper [Vilnis & McCallum, 2014] which makes appear the paper a minor improvement of what it is actually.

---

> ### Author Response · Authors · 2017-12-29
> **Response to AnonReviewer3**
>
> Thank you for your thoughtful comments and your interest in the paper. Please see our responses below.
>
> [Clarification on Introduction]
>
> Indeed the scarcity of lexical relationships in natural text corpus is not the problem we aim to solve in this paper. We brought this up to emphasize the importance of modeling hierarchical data via *supervised* learning directly on hierarchical data, without relying on word occurrence patterns in the unsupervised approach. Note that in the unsupervised setting, the Gaussian distributions are trained based on word occurrences in natural sentences. In the supervised setting, we model the hierarchical data by minimizing the loss directly on relationship pairs.
>
> The supervised learning of Gaussian densities has not been thoroughly considered in the existing literature. The main goal of our paper is to investigate this highly consequential task. Vilnis (2015) proposed a Gaussian embedding model that works well for unsupervised task (semantic word embeddings). While the approach can be directly applied to supervised case, the performance is often quite poor based on results reported in Vendrov (2015) and Vuli\'c (2016) . Our paper, on the other hand, shows the opposite finding that the performance of Gaussian embeddings can be highly competitive so long as we use our different new training approach. We would like to emphasize the significance of this direction: despite their intuitive benefits in providing rich representations for words, probabilistic word embeddings are relatively unexplored.  Such embeddings are only considered in a small handful of papers, and in these papers there is no serious consideration of where these embeddings would be most natural, such as in ordered representations.
>
> We would also like to emphasize that we do introduce new methodology in our paper. We propose new training procedures that learn highly effective Gaussian embeddings in the supervised setting. The new changes include (1) using max-margin loss (Equation 7) instead of rank loss (Equation 1); (2) using a divergence threshold to induce soft partial orders and prevent unnecessary penalization; (3) a new scheme to select negative samples for max-margin loss; (4) investigating other hyperparameters that are important for density encapsulation such as adjusting variance scale; (5) proposing and investigating general alpha divergences as metrics between word densities.  This direction is significant — analogous to exploring Renyi divergences instead of variational KL divergences for approximate Bayesian inference.
> We thank the reviewer for the questions/comments and we have modified the introduction to make our contributions more clear.
>
> [Hyperlex] HyperLex is an evaluation dataset where the instances in HyperLex have some lexical relationships. Instances that have true hypernym relationships have high scores, and instances
> without true hypernym relationships have lower scores.
>
> [Test Set] 4000 Hypernym pairs are selected randomly from the transitive closure of WordNet. The random split is 4000 for validation set, 4000 for test set, and the rest for training.
>
> [Diagonal Covariance] The diagonal covariance enables fast training. The complexity of the objective for
> d-dimensional is O(d) for diagonal case as opposed to O(d^3) in the full covariance case.
> This is due to the inverse term in our divergences. Note we use a full diagonal covariance versus a scaled identity. Relative to standard word embeddings, such as word2vec, a probabilistic density, even with a diagonal covariance, is highly flexible.
>
> [Figure 4] Yes, we can see that ‘location’ and ‘living_thing’ are visually overlapping. However,
> KL(‘location’ | ‘living_thing’) = 6794 and KL(‘living_thing’ | ‘location’) = 4324, which is higher than the threshold of 1900 (picked using validation set), which means that we do not predict that these words entail one another in either direction. There are some mistakes, however, such as for ‘location’ | ‘object’, which can happen when there are not enough negative examples of the pair (‘location’ (not) > ‘object’) to contrast in the training.  Our proposed negative sampling approach helps alleviate this problem.

---

### Official Review · AnonReviewer1 · 2017-11-28

**Rating:** 8
**Confidence:** 5

**Review:**

The paper introduces a novel method for modeling hierarchical data. The work builds on previous approaches, such as Vilnis and McCallum's Word2Gauss and Vendrov's Order Embeddings, to establish a partial order over probability densities via encapsulation, which allows it to model hierarchical information. The aim is to learn embeddings from supervised structured data, such as WordNet. The work also investigates various schemes for selecting negative samples. The evaluation consists of hypernym detection on WordNet and graded lexical entailment, in the shape of HyperLex. This is good work: it is well written, the experiments are thorough and the proposed method is original and works well.

Section 3 could use some more signposting. Especially for 3.3 it would be good to explain (either at the beginning of section 3, or the beginning of section 3.3) why these measures matter and what is going to be done with them.

It's good that LEAR is mentioned and compared against, even though it was very recently published. Please do note that the authors' names are misspelled: Vuli\'c not Vulic, Mrk\v{s}i\'c instead of Mrksic.

If I am not mistaken, the Vendrov WordNet test set is a set of positive pairs. I would like to see more details on how the evaluation is done here: presumably, the lower I set the threshold, the higher my score? Or am I missing something?

It would be useful to describe exactly the extent to which supervision is used - the method only needs positive and negative links, and does not require any additional order information (i.e., WordNet strictly contains more information than what is being used).

I don't see what Socher et al. (2013) has to do with the loss in equation (7). Or did they invent the margin loss?

Word2gauss also evaluates on similarity and relatedness datasets. Did you consider doing that here too?

"hypothesis proposed by Santus et al. which says" is not a valid reference.

---

> ### Author Response · Authors · 2017-12-29
> **Response to AnonReviewer1**
>
> Thank you for your thoughtful and supportive comments. Please see our responses below.
>
> [Misspelled names and reference errors] Thank you for pointing this out. We have corrected the spelling and reference errors.
>
> [Equation 7] Socher et al. 2013 and Vendrov 2015 uses this loss in their tasks.
>
> [Effects of Divergence Threshold]
>
> Vendrov’s WordNet test set is a test of 4000 positive pairs as well as 4000 negative pairs, where the negative pairs are selected using random sampling. We fix the same test set across all experiments.
>
> In the Appendix, Section A.2.1, we show the effects of the divergence threshold (gamma) on the test accuracy. Figure 5 shows that there is an optimal gamma value which yields the best scores on hypernym prediction and lexical entailment.The intuition is as follows: zero gamma corresponds to penalizing any two distributions that do not perfectly overlap (since D is the lowest if and only if the two distributions are equal). This behaviour is undesirable: if a distribution is correctly encapsulated by a parent distribution we should not further penalize the embeddings. High gamma corresponds to low penalization among many distribution pairs — a gamma value that is “too high” is lenient, since there might not be sufficient penalization in the loss function to help learn optimal embeddings.
>
> [Similarity and Relatedness]
>
> We believe that similarity and relatedness would be more suitable for word embeddings trained on word occurrences in a natural corpus (word2vec, GloVe, word2gauss), because our embeddings model the hierarchical structure rather than the semantics of concepts. In Section 5, we discussed the future direction where we plan to use the idea to enhance word embeddings (Gaussian word distributions) by incorporating the supervised training (using labels from WordNet, for instance) with the unsupervised training on text corpus. The ideal scenario would be that the Gaussian embeddings would have high word similarity scores and also exhibit hierarchical structure that yields good hypernym prediction and graded lexical entailment scores.

---

### Author Response · Authors · 2018-01-04
**Changes in New Draft**

We have incorporated the reviewers' comment into our new draft. The changes are as follow:

Section 1. Introduction
We further described the distinction between our work and existing work and made our contributions more explicit.

Section 4. Experiments
Section 4.1 We added more explanation on the hypernym prediction test set.
Section 4.2 We explained why ELK can work reasonably well on hypernym prediction.
Section 4.4 We elaborated on the negative sample selection and discussed the results from using different combinations. We also added the result from using a symmetric model to Table 3.

In addition, we slightly changed the title and added additional keywords to facilitate better search for our paper.

---

### Public Comment · ~Luke_Vilnis1 · 2018-01-31
**New paper title and relevant prior work**

I read this interesting work (I especially enjoyed learning about the idea of the proposed strict "set measure") when it was first posted under its original title, "ON MODELING HIERARCHICAL DATA VIA ENCAPSULATION OF PROBABILITY DENSITIES." I think think this was a more accurate title considering, as the authors note, the proposed model doesn't necessarily obey transitivity, nor is it a probabilistic model of a transitive relation -- the cutoffs are on KL divergences, not edge probabilities. Instead, the embeddings themselves take the functional form of abstract probability distributions, like our original Gaussian work.

A missing reference is Lai and Hockenmaier's EACL 2017 paper "Learning to Predict Denotational Probabilities For Modeling Entailment"  (https://www.aclweb.org/anthology/E/E17/E17-1068.pdf), which does directly learn a transitive, probabilistic order embedding in the sense that if P(mammal|dog) approx 1 (so mammal < dog in the ordering with probability 1), and P(animal|mammal) approx 1  (so animal < mammal in the ordering), then P(animal | dog) approx 1 ( animal < dog in the ordering ) is assured. With lower cutoffs than P close to 1 the exact bound depends on depth, dimensionality, etc.

They do this by taking the order embedding model and assigning an exponential measure to the conic sublattices implied by each vector under the ordering relation, defining the probability of P(x) as exp(-sum_i x_i) and P(x,y) as exp(-sum_i max(x_i,y_i)), and the edge probability from there as conditional probability of parent given child.

I encourage the authors to take a look at that work and cite it in the camera-ready, as it was initially what sprung to mind when I heard "probabilistic order embeddings" and is quite different.

After mentioning someone else's work, on a more selfish note, on the subject of picking graph-aware training examples for order embeddings, the authors might also be interested in Xiang Li's and my ICML 2017 workshop paper, "Improved Representation Learning for Predicting Commonsense Ontologies" (https://arxiv.org/abs/1708.00549), where we get an improvement from 90.6 to 91.3 on standard order embeddings just by including extra constraints based on ancestors using the join and meet operations of the lattice (Table 3).

Thanks for your time!

---

### Decision · Program_Chairs · 2018-01-29
**ICLR 2018 Conference Acceptance Decision**

**Decision:**

Accept (Poster)

**Comment:**

This paper marries the idea of Gaussian word embeddings and order embeddings, by imposing order among probabilistic word embeddings. Two reviewers vote for acceptance, and one finds the novelty of the paper incremental. The reviewer stuck to this view even after rebuttal, however, acknowledges the improvement in results. The AC read the paper, and agrees that the novelty is somewhat limited, however, the idea is still quite interesting, and the results are promising. The AC was missing more experiments on other tasks originally presented by Vendrov et al. Overall, this paper is slightly over the bar.